# Fruit and Vegetable Lesson Plan Pilot Intervention for Grade 5 Students from Southwestern Ontario

**DOI:** 10.3390/ijerph17228422

**Published:** 2020-11-13

**Authors:** Sarah J. Woodruff, Clinton Beckford, Stephanie Segave

**Affiliations:** 1Department of Kinesiology, University of Windsor, Windsor, ON N9B 3P4, Canada; 2Faculty of Education, University of Windsor, Windsor, ON N9B 3P4, Canada; clinton@uwindsor.ca; 3Ontario Student Nutrition Program-Southwestern, Windsor, ON N8W 5C2, Canada; stephanie.segave@von.ca

**Keywords:** fruit and vegetables, nutrition intervention, lesson plans, school health

## Abstract

The purpose was to create and assess the impact of food literacy curriculum alongside a centrally procured school snack program among grade five students in Southwestern Ontario, Canada. Grade five students (*N* = 287) from five intervention and three controls schools participated in an 8-week food delivery program. In addition to the food delivery program, intervention schools received a resource kit and access to 42 multidisciplinary food literacy lesson plans using the produce delivered as part of the food delivery program. Participants completed matched pre- and post-test online surveys to assess fruit and vegetable intake, knowledge, preferences, and attitudes. Descriptive analyses and changes in scores between the intervention and control schools were assessed using one-way ANOVAs, paired samples *t*-tests, and McNemar’s tests. In total, there were 220 participants that completed both the pre- and post-test surveys. There was a significant improvement in fruit and vegetable intake (*p* = 0.038), yet no differences in knowledge of the recommended number of food group servings, knowledge of food groups, or fruit and vegetable preferences or attitudes were observed. Integrating nutrition lesson plans within core curricula classes (e.g., math, science, and literacy) can lead to modest increases in fruit and vegetable intake.

## 1. Introduction

Despite Canadian children consuming adequate carbohydrates, protein, and fat as recommended [1], fruit and vegetable (FV) intake among school-aged children and adolescents is concerning. Data from the Canadian Community Health Survey in 2017 [2] suggested that only 30.7% of males and females aged 12–17 years consumed five or more servings of FV per day. Further, among a large Canadian study of students in grades 6-12 from nine provinces (*N* = 47,203), only 9.9% self-reported meeting the previous recommendations of Canada’s Food Guide [3]. An important source of vitamins and minerals and fiber, FV are essential for proper growth and development [4]. Often used as a proxy for healthy eating, FV consumption has been associated with various health outcomes in both the short- and long-term [5,6,7]. Increasing FV consumption has been a central feature of public health initiatives to promote healthy eating over the last few decades.

School food programs (i.e., breakfast, lunch, and/or snack programs) offer a promising approach to support and teach about healthy food behaviors, which may provide lifelong health benefits. For example, food preferences and liking new foods has been associated with increased exposure to healthy food via school provisions [8,9,10], and increased preference for FV has been associated with longitudinal FV intake from adolescence to young adulthood [11]. A meta-analysis mainly consisting of studies from the United States and Europe suggested that moderate fruit intake increases were associated with school-based FV interventions, even though they had no impact on vegetable intake [12].

Canada is the only G7 country (i.e., highly industrialized) without a national school nutrition program. Various regional or provincial models exist to provide snack programs to schools, yet there is no consistency among programs. Furthermore, there is a lack of evidence within Canada to assess the benefits of the programs. A recent systematic review of elementary school nutrition program evaluations in Canada resulted in only nine school-based programs being identified that assessed the impacts on children’s nutrition [13]. Most of those nine programs were multidimensional (i.e., including policy, education, family and community involvement, and/or food provisions), and resulted in improvements in nutrition knowledge, dietary behaviors, and/or food intake. However, great variability between these programs existed and a lack of reproducibility for outcomes warrants further investigation into school snack programs in Canada.

More broadly, research also suggests that school-based interventions that include an educational component (e.g., healthy eating resources, food skills, and gardening) can be effective for enhancing student’s nutritional knowledge and FV intake [8,13,14]. However, a recent systematic review among the associations between food literacy (i.e., food knowledge and frequent food preparation) and dietary intake among adolescents suggested mixed results even though the majority of studies reported a positive association between food literacy and healthier dietary practices [15]. Therefore, determining how best to implement a food literacy component to an existing school-based nutrition program seems like a worthy effort.

In Ontario, Canada, the Ministry of Children, Community, and Social Services provides 14 regional lead agencies with funding to work in partnership with local communities to flow seed grants and support to locally delivered breakfast, lunch, and/or snack programs. In Southwestern Ontario, many schools participate in a centrally procured snack program (i.e., food delivery program), whereby students receive one serving of either a fruit or vegetable each day for eight consecutive weeks via weekly deliveries to the schools. To date, no educational components/food literacy has supported this snack program even though it was delivered to just over 150 schools (46,000 students) in the 2016–2017 academic year. Therefore, the purpose of this study was to create and assess the impact of food literacy lesson plans, which would be delivered in conjunction with a centrally procured school snack program (one fruit or vegetable, 5 days/week, for 8 weeks). Specifically, FV intake, preferences, knowledge, and attitudes were assessed among grade five students to determine the impact of a food literacy curriculum that was delivered in conjunction with an ongoing food delivery program. It was hypothesized that participants in the intervention classrooms (i.e., receiving the food literacy lesson plans) would have better outcomes compared to participants in classrooms with only the food delivery program.

## 2. Materials and Methods

Participants and Recruitment: Grade five students from five intervention and three control schools were invited to participate. Information letters and consent forms were sent home asking parents to provide active consent. Only those with parental consent completed the pre- and post-test survey. Participants provided assent upon starting the online survey. In total, there were 287 potential participants in grade five (*n* = 139 intervention and *n* = 148 control). All data were collected in the spring of 2018 and all procedures were approved by the University of Windsor Research Ethics Board, the participating school board, and principals of the participating schools.

### 2.1. Instruments

Pre- and post-test online surveys were completed. All questions were taken from validated surveys and/or national surveys [10,16,17,18,19]. Participants were asked to create an anonymous identification code (based on their birthday and last 2 letters of their last name) in order to track participants from pre- to post-test. The survey asked about FV intake, preferences, knowledge, and attitudes, in addition to demographics (pre-test) and questions about the snack program and food literacy component (post-test). Upon completion of the surveys, one student from each school had the opportunity to win a free pass to a local water-theme park (at both pre- and post-tests). The survey took approximately 10 min to complete under the supervision of the research team.

FV intake was assessed using the question “On a usual day, how many servings of fruit and/or vegetables do you eat? (Include fresh, frozen, canned, and cooked items like apple, banana, carrot, salads, and 100% juice. Do not include chips, French fries, or other fried potatoes)” [3]. Examples of single servings specified “1/2 cup of fresh, frozen, or cooked vegetables”, “1 cup of raw leafy vegetables; like a small salad”, “1 medium fruit; like an apple, pear or banana”, and “1/2 cup of 100% fruit or vegetable juice”. Response options included: 0 servings; 1–2 servings (used 2 servings for statistical purposes); 3–4 servings (used 4 servings for statistical purposes); 5 servings; 6 servings; 7 servings; and 8 or more servings.

FV preferences were assessed separately using a 5 point Likert-type scale (happy faces ranging from very smiley to unhappy; with options for “have never tried/don’t know” and “I am allergic”) for 10 fruit and 10 vegetables [10,20]. Within the examples of FV, seven were offered as part of the FV snack delivery program (and three were randomly chosen). Each FV was given a score out of 4 (0 = never tried and a range of 1 = unhappy face to 4 = very smiley face; if allergic, the total was pro-rated) and then averaged.

In order to assess the participants knowledge of Canada’s Food Guide, participants were asked “How many servings of fruit and vegetables (for example 1 whole fruit, ½ cup frozen/canned vegetables, 1 cup raw salad) do you think you should eat every day to stay healthy?” with response options of 5 servings; 6 servings; 7 servings; and 8 servings [20]. Responses were scored as correct (1) or incorrect (0). Knowledge of food groups was assessed by asking participants to identify whether a specific food (i.e., rice, banana, yogurt, cheese, celery, chicken, almonds, bread, and carrots) was a fruit, vegetable, or other food [21,22]. All correct scores were assigned 1 and scores were summed (out of nine) and converted to a percentage of correct answers.

FV attitudes was assessed using an 11-part question asking “What do you think about eating fruit and vegetables?” with example statements like “I think fruit tastes good” and “I will have more energy if I eat fruit and vegetables” [23]. Responses were scored using a 5 point Likert-type scale (happy faces ranging from very smiley = 4 to unhappy = 1; with an option for “have never tried/don’t know” = 0) and summed with higher scores representing higher FV attitudes.

### 2.2. Procedures

#### 2.2.1. Intervention

Teacher candidates from a non-mandatory service-learning course in the Faculty of Education at the University of Windsor created a series of food literacy lesson plans (*n* = 42) using the weekly food delivery menu and the grade five Ontario curriculum. The lesson plans targeted drama, health and physical education, science, English writing, language arts, social studies, French, math, and visual arts. All lesson plans were reviewed for accuracy and completeness by a Registered Dietitian. Some additional resources (i.e., growing pots and soil, markers/paint, recipe cards, playing cards, bowl, wooden sticks, etc.) were purchased for several of the lesson plans to include as part of a curriculum resource kit (i.e., lesson plans and resources). The resource kit, in addition to access to a shared online drive with the lesson plans, was delivered to five intervention schools (eight classrooms) for the grade five teachers. Teachers were instructed to use any/all of the lesson plans over the next eight weeks in conjunction with the school snack program.

#### 2.2.2. FV Delivery Program

All schools within one region in Southwestern Ontario received one FV per day (per student) for eight weeks. The pre-set menu used FV that were centrally procured (e.g., purchased in bulk) and then directly delivered to the schools. Although potentially limited by the Ontario growing season, the program had a set target of no less than 20% of foods offered in any week from local sources, although in most weeks it was much higher. As opposed to traditional student nutrition programs (whereby volunteers purchase food from retailers with little to no control over what items are purchased), this program ensures that foods offered are of a consistently high nutrient quality and are guaranteed to follow the Ontario Student Nutrition Program Guidelines set forth by the Ministry of Children and Youth Services [24].

### 2.3. Data Analysis

Descriptive results are reported as mean ± standard deviation (SD) or percentage. To assess differences in FV intake, FV preferences, knowledge of food groups, and FV attitudes, separate paired samples *t*-tests were computed based on differences between the pre- and post-test scores between the intervention and control conditions. A McNemar’s test was used to determine changes in correctly identifying the recommended number of FV servings from Canada’s Food Guide between pre- and post-test by intervention and control conditions. All statistical procedures were computed using Minitab 17.3.1 (State College, PA, USA) using a level of significance of 0.05.

## 3. Results

In total, 273 participants (95% response rate) completed either the pre- or post-test: 242 participants completed the pre-test (84% response rate) and 249 completed the post-test (87% response rate). There were 220 participants that had completed the surveys at both times (i.e., matched data). Table 1 provides the raw data from all participants at the pre-test, suggesting that there were no statistical differences in any of the demographic or measured outcome variables between the intervention or control group (*p* > 0.05), with the exception of FV attitudes being higher in the intervention group.

All data for the matched participants (*n* = 220) are presented in Table 2. A paired samples *t*-test was conducted to compare the changes in FV intake in the intervention and control conditions. Overall, there was a significant difference between the conditions for FV intake changes, T (206) = −2.08, *p* = 0.038, such that greater improvements were observed among those in the intervention (i.e., intervention participants consumed 0.25 servings more at the post-test than pre-test). A paired samples *t*-test was conducted to compare the effect of fruit preference changes in the intervention and control conditions and suggested that there was no significant effect of condition on fruit preference changes, T (185) = −1.82, *p* = 0.071. Further, a paired samples *t*-test was conducted to compare the effect of vegetable preference changes in the intervention and control conditions such that was no significant effect of intervention on vegetable preference changes, T (181) = −1.59, *p* = 0.114.

A McNemar’s test, conducted to compare the effect of correctly identifying the recommended number of servings from Canada’s Food Guide at post-test (compared to pre-test) within each condition, suggested there was no difference in changes within the intervention (*p* = 0.152) or control conditions (*p* = 0.644). Further, a paired samples *t*-test was conducted to compare the changes in knowledge of food groups based on intervention/control condition suggested that there was no significant effect of intervention on knowledge of food groups, T (192) = 0.85, *p* = 0.395. Lastly, paired samples *t*-test to compared the changes in FV attitude changes in the intervention and control conditions suggested that there was no significant effect of intervention on FV attitude changes, T (162) = 0.93, *p* = 0.355.

## 4. Discussion

This study investigated the impact of a teacher led food literacy curriculum intervention alongside an ongoing centrally procured school snack program among grade five students from Southwestern Ontario, Canada. The 42 food literacy lesson plans were designed so that they spanned nine different subject areas (in lieu of just health and physical education). Overall, FV intake among the current participants was lower than the six servings (for 9–13 year olds) as recommended by Canada’s Food Guide, even though all participants received the food delivery program. The current FV intake (with one serving being provided during the school day) is in line with other Canadian studies [3,8,25,26], suggesting the need for further health promotion and public health efforts. Furthermore, the data suggest that there was a significant impact of the food literacy curriculum for FV intake, while no effects were observed for knowledge of the recommended number of food group servings, knowledge of food groups, FV preferences, or attitudes.

Overall, FV intake increased at post-test by 0.25 servings among the intervention (food literacy curriculum with delivery program) compared to the control (food delivery program only) participants. Many schools in Southwestern Ontario receive the food delivery program, therefore, it is unknown what the intake would be like without any intervention at all, albeit likely lower. However, the modest increase between the two conditions is in line with reviews and meta-analyses suggesting usual increases of less than 1 serving of FV/day for school-based nutrition interventions [12,27,28,29]. Interestingly, a study from Northern Ontario suggested modest increases in a food delivery program compared to a control group, even though no differences were observed with or without educational supports within the two arms of the intervention [8]; however, the authors reported that the educational component was not fully implemented as planned [30]. Among Canadian studies, over half of the programs assessed in the recent systematic review of school nutrition programs [13] reported increased consumption of healthy foods, including FV [25,31,32] and milk and alternatives [9], therefore, it could be thought that the addition of education supports (such as lesson plans) slightly improved intakes beyond no intervention at all. Yet, the FV increase observed in the intervention group could also be due to the education in helping them to better assess the serving amounts rather than them changing their behavior. However, given the myriad of potential influences on food intake and that the one serving of FV was provided during school hours through this FV program to all students, further research is necessary to determine how to increase FV intake to meet recommended standards.

The present findings also suggest that the food literacy lesson plans did not improve knowledge of Canada’s Food Guide recommended number of FV servings, knowledge of food groups, FV preferences, or attitudes. Yet, it is important to note that the scores for correctly identifying a food into its food group and FV attitude scores were already fairly high at baseline for both conditions, and it is likely that ceiling effects may have occurred. As reported in the recent systematic literature review of Canadian school food programs [13], while the majority of the school food program studies showed promising/positive effects on food knowledge, preferences, and attitude scores not all results are consistent across the different types of programs. Furthermore, the food literacy curriculum developed for the current study did not specifically target these measured outcomes but were designed to meet learning goals for other subjects (i.e., drama, health and physical education, science, English writing, language arts, social studies, French, math, and visual arts) while exposing them to different FV’s as their medium or example. It is also possible that although 42 different food literacy lesson plans were created, it was up to the teacher to use and implement as they saw appropriate. All intervention teachers reported using several of the lesson plans (data not shown), however, no implementation evaluation occurred, thereby not knowing how well the lesson plans were delivered (i.e., fidelity) and/or accepted by the students.

### Practical Applications and Limitations

A major outcome of this research is the creation of the food literacy lesson plans spanning nine different subject areas, as most notably, nutrition education interventions generally only target healthy eating and/or the health and physical education curriculum [9,25,33,34]. It has been reported that elementary school teachers often display low levels of nutrition knowledge, self-efficacy and skills to deliver nutrition education [35] and using multiple curricular approaches in different subject areas may help overcome teacher concerns with specific nutrition knowledge. Furthermore, many studies utilize a teacher training module (i.e., anywhere from one hour to several days) prior to the intervention to educate and acquaint the teachers with the material. However, demand for teacher time outside of the classroom is nearly impossible to acquire and the current strategy of simply providing the materials was agreed upon by the partnering school board. Lastly, using the teacher candidates (teachers in training) to create the lesson plan provided them with hands on learning of how to apply different curriculums with health outcomes and created lesson plans for them to use in the future. Despite these strengths, however, several limitations exist. Data were self-report, based on survey questions taken from other validated surveys, which has implications for reporting bias, errors, and memory recall. Furthermore, this study included only five intervention schools (eight teachers) and not the entire geographic region or school board, thus may not be representative of the larger population. Finally, this study only included grade five students, and future research and educational initiatives should expand and include other grades to determine differences that potentially occur with age.

## 5. Conclusions

In sum, the food literacy lesson plan intervention resulted in slight increases in FV intake, with no significant differences observed for knowledge of the recommended number of FV servings from Canada’s Food Guide, knowledge of food groups, FV preferences, or FV attitudes. The present data supports using the integration of health education efforts within core curricula classes (e.g., math, science, and literacy) among elementary students. For example, future research may want to investigate their implementation in a more systematic and controlled manner, however, the manner in which they were available for the teachers in the present study is fairly indicative of “real world” teaching. Other researchers have noted that curriculum education has not always gone as planned [29,34,36,37] as often there are competing academic interests, unintended interruptions (e.g., snow days and field trips), and/or a lack of consistent lesson fidelity across teachers. For example, between two primary school-based interventions, compared to just FV distribution, the multicomponent program (i.e., age-specific program of classroom curriculum, parental involvement, and environmental components) was less implemented and activities decreased over time even though both interventions were rated favorably [37]. Practically, the present results indicated modest increases, which could be expanded to more grades and/or other provincial/state curriculums. In regard to research, a longer duration evaluation (i.e., follow-up) is needed and/or longitudinal studies to determine the impact of school-based nutrition programs on health outcomes in Canada. Further, more studies of integrated approaches that incorporate several components to support student nutrition and healthy eating habits is necessary.

## Figures and Tables

**Table 1 ijerph-17-08422-t001:** Participant Demographics and Fruit and Vegetable intake, Preferences, Knowledge, and Attitude Scores at Baseline (*N* = 273).

	Intervention (*n* = 139)	Control (*n* = 148)	Difference between Intervention and Control ^1^
Age (years)	10.2 (0.5)	10.2 (0.4)	0.696
Gender (% male)	51%	49%	0.796
Ethnicity (% Caucasian)	52%	47%	0.773
Fruit and vegetable intake (servings/day; M (SD))	4.3 (1.7)	4.4 (2.1)	0.908
Fruit Preferences (scores range from 0-4; M (SD))	2.8 (0.7)	2.8 (0.9)	0.747
Vegetable Preferences (scores range from 0-4; M (SD))	2.3 (0.8)	2.1 (0.9)	0.122
Knowledge of food groups (maximum score = 9; M (SD))	8.4 (1.1)	8.4 (1.3)	0.824
Knowledge of recommended number of servings (% correct)	35%	31%	>0.05 ^2^
Fruit and vegetable attitudes (maximum score = 44; M (SD))	35.1 (4.2)	32.9 (6.5)	0.003

Mean (M), standard deviation (SD). ^1^ one-way ANOVA to assess differences between conditions. ^2^ based on separate McNemar tests for between group differences.

**Table 2 ijerph-17-08422-t002:** Changes in Fruit and Vegetable Intake, Preferences, Knowledge, and Attitudes Scores between Pre- and Post-test (*n* = 220 matched participants).

	Intervention (*n* = 13)	Control (*n* = 107)	Difference between Intervention and Control ^1^
Fruit and vegetable intake (servings/day; M (SD))	0.25 (1.9)	−0.31 (2.0)	0.038
Fruit Preferences (scores range from 0-4; M (SD))	0.10 (0.4)	−0.27 (0.5)	0.071
Vegetable Preferences (scores range from 0-4; M (SD))	0.07 (0.4)	−0.03 (0.5)	0.114
Knowledge of food groups (maximum score = 9; M (SD))	−0.13 (0.86)	−0.021 (0.92)	0.395
Knowledge of recommended number of servings (% correct)	43.7%	27.1%	>0.05 ^2^
Fruit and vegetable attitudes (maximum score = 44; M (SD))	0.29 (3.7)	0.89 (4.7)	0.355

Mean (M), standard deviation (SD). ^1^ paired t-test of the difference between pre- and post-tests by intervention or control. ^2^ based on separate McNemar tests for within group differences.

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
