# Peer review of "Fruit and Vegetable Lesson Plan Pilot Intervention for Grade 5 Students from Southwestern Ontario"

_ijerph, 2020, doi:10.3390/ijerph17228422_

Round 1

Reviewer 1 Report

Review of

Fruit and vegetable lesson plan intervention for 3 Grade 5 students from Southwestern Ontario

Line 97 …. “Participants were asked to create an anonymous 93 identification code in order to track participants from pre- to post-test.”  This seems problematic for a 5th grader to remember what code they used

Do 5th graders know serving sizes to be able to estimate their intake. Pictures of servings have been shown to be helpful.

Even though you took your questions from validated surveys- they were from a variety of sources and might not have been intended for 5th graders. How did you validate your final survey? What type of pre-test did you do?

What are the dangers of assessing F/V intake using one global question? It would have been better to have Fruits and vegetables assessed separately. Most adults have a very hard time assessing their own intake . When you provided F/V their immediate  intake would increase—how are you affecting their behaviors at home?  How did you pick your attitude questions – are they based upon a theoretical framework such that if you influence these attitudes it should affect x behavior?

OK the lesson plans were reviewed by an RD but how were they pilot tested with the intended audience and 5th grade teachers?

You should have assessed what the lessons the teachers used to really know your intervention dose for the education.

Your results --increased intake might actually be due to the education helping them to better assess their amounts rather than them actually changing their intake.

Data analysis – how can you be sure they are appropriately matched? How did you clean the data?  

Why are you reporting # pre and # post if they are truly matched surveys? You really only have a set of 53  matched surveys ??  This is misleading in the abstract and needs to be fixed. Is this a sufficient sample size for your analysis?

None of these analysis are based upon matched data …you are compare the two groups pre/post

The paragraph on the top of page 7 Line 252 to 264 is interesting but not studied/assessed or supported by your results.

I would call this paper a pilot test and would go back and do all of the previously suggested assessments (pilot-tests of curriculum/tools) clean up anonymous ID, etc.  before actually running and reporting the intervention study.

Author Response

Line 97 …. “Participants were asked to create an anonymous 93 identification code in order to track participants from pre- to post-test.”  This seems problematic for a 5th grader to remember what code they used

This has been clarified on line 89.  The code was based on their birthday and last 2 letters of their last name, so was an easy code to remember that was anonymous to the researchers.

Do 5th graders know serving sizes to be able to estimate their intake. Pictures of servings have been shown to be helpful.

This question has been used in large Canadian studies of children/adolescents to measure FV intake. Although no pictures were present, the question does provide examples which are stated on Lines 97-99.

Even though you took your questions from validated surveys- they were from a variety of sources and might not have been intended for 5th graders. How did you validate your final survey? What type of pre-test did you do?

There are no well-known Canadian validated nutrition surveys that were specific to our population/topic.  As such, we chose questions from other surveys that have been used within the Canadian context among this age group in order to be able to make comparisons from our sample to others.  I have personally used and published with the majority of these questions for several similar studies.

What are the dangers of assessing F/V intake using one global question? It would have been better to have Fruits and vegetables assessed separately. Most adults have a very hard time assessing their own intake . When you provided F/V their immediate  intake would increase—how are you affecting their behaviors at home?  How did you pick your attitude questions – are they based upon a theoretical framework such that if you influence these attitudes it should affect x behavior?

We acknowledge that it might have been better to assess FV in two separate questions, but we chose the best question available to us at the time of the study design.  This has now been mentioned in the limitations on lines 272-273. The FV attitudes question was a more global question about how they feel about FV. We thought since we were including FV into other areas of study (not just Health/PE class) that they might change their attitudes (but this was not observed).  Lastly, we did not consider the home/parent environment (even though the daily FV intake would have accounted for it it).

OK the lesson plans were reviewed by an RD but how were they pilot tested with the intended audience and 5th grade teachers?

As you mention in another comment below, perhaps the best manner in moving this study forward is by calling a pilot study – which has now been changed in the title.  We did not pilot test with the intended audience or the teachers (nor where they trained, as explained in the paragraph on lines 254-272).  The decision not to pilot with a small group of students/teachers is really about the lack of time and availability that students/teachers have to give to research projects such as this and our agreement with school board for allowing us permission to conduct the study. 

You should have assessed what the lessons the teachers used to really know your intervention dose for the education.

As mentioned on line 251, teachers provided feedback re using several of the lesson plans, but not specifically which one’s they used or didn’t use.  Along the same lines as the previous response (and as mentioned in paragraph on lines 258-277), and while not measured directly in this study, this is the reality of teachers within our sampling area. Access to teachers is near impossible and the teachers appreciated the flexibility in the use of the intervention.  Informal feedback suggested that they may not have participated in the study in the first place had there been strict requirements to try so many of the lesson plans.

Your results --increased intake might actually be due to the education helping them to better assess their amounts rather than them actually changing their intake.

This is a good point, and is now alluded to in lines 236-237.

Data analysis – how can you be sure they are appropriately matched? How did you clean the data?  

Given this comment, as well as others, the data cleaning, matching, and demographics have been updated (lines 159-165, plus Table 1).

Why are you reporting # pre and # post if they are truly matched surveys? You really only have a set of 53  matched surveys ??  This is misleading in the abstract and needs to be fixed. Is this a sufficient sample size for your analysis?

The abstract has now been updated to mention the 220 matched participants (line 19). The additional changes to Table 1 and Table 2 should make things clearer.

None of these analysis are based upon matched data …you are compare the two groups pre/post

This comment, and Reviewer #2’s comment, regarding a similar issue has now been updated.  Table 1 includes all data (unmatched to show there was no differences between the intervention and control groups at pre-test), and table 2 (and all the analyses) are based on matched participants.

The paragraph on the top of page 7 Line 252 to 264 is interesting but not studied/assessed or supported by your results.

Respectfully, we have kept this text in the document as we feel it contextualizes some of the results within the Canadian context. However, we have added the title of “Practical Applications and Limitations” before this section to separate it out from the discussion.

I would call this paper a pilot test and would go back and do all of the previously suggested assessments (pilot-tests of curriculum/tools) clean up anonymous ID, etc.  before actually running and reporting the intervention study.

Your suggestion to report this as a pilot study has been noted in the title.

Reviewer 2 Report

This is a timely intervention and lot of activities are involved. Very useful area for readers. Well done intervention. Method was well written. 

Introduction should be improved to show more on Canadian aspect with data. Fruit and vegetable consumption data of children and may be the importance of it with NCD pattern of the country. Small paragraph. 

Need following revisions:

Line 30 – remove bracket with fruits

Line 34 – previous recommendation delete (then) - previous guide

Line 160-164 – Amalgamate - same thing repeating twice. you can add both factors in the same sentence. " A one-way ANOVA was conducted to compare the changes (matched from pre- to post-test) in knowledge of food groups and to assess FV attitudes, to compare the changes (matched from pre- to post-test) based on intervention/control condition".

Line 180-187 –Table 1 – need the statistical calculation in each control and intervention separately. It was observed in control group also, there is a change between pre and post test, better to look for any statistical change. Some variables are mean some are %. Better to label the data correctly under each variable. Most appropriate statistical test will be paired t test and McNemar test. Apply Paired T test and McNemar test for within groups. Need to show the change in Each group and then compare the change with each group, Table 1 should be expanded to show all these areas.

Better to add the basic table as Table 1 to show findings of each component, which is described in the text but no data.

Line 196-198 need to show data for this explanation

Line 275-277 – Difficult to express from this study, better to delete

Some of the points in the discussion not shown in the results section, need to show data. (line 249-251) 

Results section should be expanded and discussion to be changed accordingly.

Author Response

This is a timely intervention and lot of activities are involved. Very useful area for readers. Well done intervention. Method was well written. 

Thank you for your positive feedback.

Introduction should be improved to show more on Canadian aspect with data. Fruit and vegetable consumption data of children and may be the importance of it with NCD pattern of the country. Small paragraph. 

Respectively, all paragraphs (but one) include Canadian data/context and we are not sure what NCD refers to.  If you could expand/clarify this point, we could try to discuss its important. However, there is a dearth of literature surround FV intake among Canadian children, which makes this data important to publish.

Need following revisions:

Line 30 – remove bracket with fruits

done

Line 34 – previous recommendation delete (then) - previous guide

“Previous” replaced “(then)”.  We also deleted “(then)” from the remainder of the paper.

Line 160-164 – Amalgamate - same thing repeating twice. you can add both factors in the same sentence. " A one-way ANOVA was conducted to compare the changes (matched from pre- to post-test) in knowledge of food groups and to assess FV attitudes, to compare the changes (matched from pre- to post-test) based on intervention/control condition".

Upon re-writing the Data Analysis section, these sentences were amalgamated with others.

Line 180-187 –Table 1 – need the statistical calculation in each control and intervention separately. It was observed in control group also, there is a change between pre and post test, better to look for any statistical change. Some variables are mean some are %. Better to label the data correctly under each variable. Most appropriate statistical test will be paired t test and McNemar test. Apply Paired T test and McNemar test for within groups. Need to show the change in Each group and then compare the change with each group, Table 1 should be expanded to show all these areas.

Better to add the basic table as Table 1 to show findings of each component, which is described in the text but no data.

Thank you for the above 2 suggestions.  A re-analysis using the appropriate measures has taken place.  Table 1 has now been divided into 2 separate tables (as per other reviewer’s suggestions to show pre-test comparisons).  As such the text in the results (and elsewhere) has been updated.

Line 196-198 need to show data for this explanation

This (never having tried a vegetable before) has been removed.  Although not indicated, never having tried a fruit was also removed.

Line 275-277 – Difficult to express from this study, better to delete

deleted

Some of the points in the discussion not shown in the results section, need to show data. (line 249-251) 

Respectively, we have kept this sentence in the manuscript as we feel it helps rationalize and explain some of our null findings.   

Results section should be expanded and discussion to be changed accordingly.

Using all of the reviewer’s comments, the results and discussion has been updated.

Reviewer 3 Report

The study fills the research gap in terms of answering the question about the matter of possible impact of a teacher led food literacy curriculum intervention among grade five students from Southwestern Ontario, Canada, who participated in an 8-week food delivery program. The article is an interesting study indicating the importance of FV consumption among 5 grade students, who are supposed to develop proper eating habits.

An extremely interesting solution to assess fruit and vegetable intake, knowledge, preferences, and attitudes among 5 grade students is the use of psychological scales regarding behaviour change, adjusted to 10-11-year-old students, both in pre- and post-test online survey. That constitutes an enormous value of this research and result interpretation. What is a great value of this article is the creation of the food literacy lesson plans spanning nine different subject areas, including: drama, health and physical education, science, English writing, language arts, social studies, French, math, and visual arts, supplemented with the resource kit. Those tools are still available and might be useful for school programmes.

However, by reading the text of the article, several questions emerge:

Abstract – is unnecessarily divided into parts, such as: (1) Background, (2) Methods, (3) Results. I checked other articles in this Journal and this way of constructing abstract is not practiced. Maybe could you omit these inclusions to maintain the layout of the abstract adopted in the journal.

One of my concerns lays with the research sample. The article completely omits the characteristics of research samples, e.g. the gender of respondents. Does this mean that it is not relevant? 5 Grade students (10-11 years old) might differ in terms of eating habits learned at home. In that time different tastes are formed among children and food preferences emerge. 5 grade students also announce more vocally that they do not want to eat certain types of products and these differences are visible in terms of gender. Was the research sample representative? Is it possible to extend the results to the other groups of students on such an attempt? What do the gender structures look like there?

Part 2 – Instruments. In my opinion this part is very unreadable and difficult to understand (lots of passages mentioned in parentheses. Could you please consider to provide a graphic/ excerpt in a graphic form from the questionnaire? Also, graph, figure showing how the author / authors combined specific features and groups of responses to obtain results would be advisable. I think this part is easy to fix. If it is not possible, please skip my comment.

The article requires some touch of editing: double spaces (e.g. line 31), unnecessary parentheses (line 30). lack of formatting (lines: 39-46).

Author Response

The study fills the research gap in terms of answering the question about the matter of possible impact of a teacher led food literacy curriculum intervention among grade five students from Southwestern Ontario, Canada, who participated in an 8-week food delivery program. The article is an interesting study indicating the importance of FV consumption among 5 grade students, who are supposed to develop proper eating habits.

Thank you for your positive feedback.

An extremely interesting solution to assess fruit and vegetable intake, knowledge, preferences, and attitudes among 5 grade students is the use of psychological scales regarding behaviour change, adjusted to 10-11-year-old students, both in pre- and post-test online survey. That constitutes an enormous value of this research and result interpretation. What is a great value of this article is the creation of the food literacy lesson plans spanning nine different subject areas, including: drama, health and physical education, science, English writing, language arts, social studies, French, math, and visual arts, supplemented with the resource kit. Those tools are still available and might be useful for school programmes.

However, by reading the text of the article, several questions emerge:

Abstract – is unnecessarily divided into parts, such as: (1) Background, (2) Methods, (3) Results. I checked other articles in this Journal and this way of constructing abstract is not practiced. Maybe could you omit these inclusions to maintain the layout of the abstract adopted in the journal.

We misunderstood the template instructions and have now removed the headings.

One of my concerns lays with the research sample. The article completely omits the characteristics of research samples, e.g. the gender of respondents. Does this mean that it is not relevant? 5 Grade students (10-11 years old) might differ in terms of eating habits learned at home. In that time different tastes are formed among children and food preferences emerge. 5 grade students also announce more vocally that they do not want to eat certain types of products and these differences are visible in terms of gender. Was the research sample representative? Is it possible to extend the results to the other groups of students on such an attempt? What do the gender structures look like there?

Table 1 has now been updated to include participant demographics.  As such, gender was ~50% male/female (and no differences between the intervention or control condition based on age, gender, or ethnicity were observed).

Part 2 – Instruments. In my opinion this part is very unreadable and difficult to understand (lots of passages mentioned in parentheses. Could you please consider to provide a graphic/ excerpt in a graphic form from the questionnaire? Also, graph, figure showing how the author / authors combined specific features and groups of responses to obtain results would be advisable. I think this part is easy to fix. If it is not possible, please skip my comment.

Unfortunately, most of the parentheses are taken directly from the question as it was asked, so we do not want to remove.

The article requires some touch of editing: double spaces (e.g. line 31), unnecessary parentheses (line 30). lack of formatting (lines: 39-46).

Again, we used the journal template to format the paper and have edited as necessary.

Round 2

Reviewer 1 Report

You explained yourself well

Reviewer 2 Report

Authors have done necessary corrections that I have mentioned in my review. I am satisfied with the corrections.